# Intent Detection and Slot Filling with Capsule Net Architectures for a Romanian Home Assistant

**DOI:** 10.3390/s21041230

**Published:** 2021-02-09

**Authors:** Anda Stoica, Tibor Kadar, Camelia Lemnaru, Rodica Potolea, Mihaela Dînşoreanu

**Affiliations:** Department of Computer Science, Technical University of Cluj-Napoca, 26-28 G. Baritiu, 400027 Cluj-Napoca, Romania; kadartibor24@gmail.com (T.K.); Rodica.Potolea@cs.utcluj.ro (R.P.); Mihaela.Dinsoreanu@cs.utcluj.ro (M.D.)

**Keywords:** NLU, intent detection, slot filling, capsule neural networks, Romanian home assistant

## Abstract

As virtual home assistants are becoming more popular, there is an emerging need for supporting languages other than English. While more wide-spread or popular languages such as Spanish, French or Hindi are already integrated into existing home assistants like Google Home or Alexa, integration of other less-known languages such as Romanian is still missing. This paper explores the problem of Natural Language Understanding (NLU) applied to a Romanian home assistant. We propose a customized capsule neural network architecture that performs intent detection and slot filling in a joint manner and we evaluate how well it handles utterances containing various levels of complexity. The capsule network model shows a significant improvement in intent detection when compared to models built using the well-known Rasa NLU tool. Through error analysis, we observe clear error patterns that occur systematically. Variability in language when expressing one intent proves to be the biggest challenge encountered by the model.

## 1. Introduction

Intent detection and slot filling are the main tasks to solve when approaching the problem of Natural Language Understanding (NLU) in a conversational system. The two tasks are used to obtain a structured representation of the meaning of the utterance, so that it can be processed by a computer. Intent detection deals with identifying the overall meaning of the sentence. It is modeled as a classification problem, in which we receive an input utterance and we have to classify it as having one intent from a group of known intents. The available intents correspond to the actions that the conversational model can perform, such as adjusting the temperature, controlling the media center or turning the lights on/off in the case of a home assistant. On the other hand, slot filling is modeled as a sequence labelling problem, whose purpose is to take the utterance and determine which words indicate relevant information for the intent. These slots contain supplementary information about the action and correspond to the parameters of the action. For instance, in the sentence *“Salut pune temperatura pe 17 grade in bucătărie”* (*“Hello set the temperature to 17 degrees in the kitchen”*) which indicates a request for setting the thermostat, the other relevant information that can be extracted is that of the value (in degrees) to which the thermostat should be set, as well as the room in which this change should be made. This information is conveyed through the sets of words *“17 grade”* (*“17 degrees”*) and *“bucătărie”* (*“kitchen”*), indicating the temperature and location information, respectively. Figure 1 illustrates the outputs produced after intent detection and slot filling for this example.

In this paper, we propose and evaluate a joint approach for intent detection and slot filling for a Romanian home assistant scenario, based on Capsule Neural Networks [2]. We consider that the input data represents user utterances in textual form. In this work, we use the words “utterance” and “sentence” interchangeably, but both are used to refer to a sentence in text format. We initially approached the home assistant scenario in [3], by generating several datasets which considered a set of language- and learning-specific challenges that are likely to occur in such a scenario: language variability (via synonymy, or other mechanisms, some specific to the Romanian language), missing values (for slots) and classification imbalance (for the various intent classes). A significant difficulty that resulted from the analysis performed in [3] was related to the misclassification of opposing intents. This was attributed to the inability of word embeddings to capture the specificity of the semantic relation between words. Additionally, missing slot values and class imbalance augmented the difficulty of the learning problem, causing a further decrease in the performance of the investigated models.

The approach we propose in this paper is based on a capsule neural network architecture that performs intent detection and slot filling simultaneously. The architecture incorporates a self-attention mechanism used to shift the attention of the model to groups of words that might be relevant for understanding the semantics of the sentence. We perform empirical evaluations aiming to study the effect of the different learning challenges on the performance of the capsule network model, as well as compare its performance to Rasa NLU models [4]. The results show that the proposed capsule network approach reaches a significantly higher intent detection performance compared to that obtained via Rasa NLU models. Additionally, the results indicate that variations in formulation when expressing one intent contribute the most to the decrease in the capsule network’s performance.

The rest of the paper is organized as follows: Section 2 discusses the main directions for solving this problem considered in the current literature. In Section 3 we present the particularities of the home assistant context, as well as details regarding the used datasets. Section 4 contains a detailed description of the proposed capsule network architecture, entitled SemCapsNet. The results of our experiments are presented in Section 5, together with detailed error analysis and an ablation study that investigates the effect of the added self-attention mechanism. Finally, in Section 6 we open new research directions to address the shortcomings of the model and Section 7 concludes the paper.

## 2. Related Work

Traditional approaches address the two learning tasks separately, generally ignoring the information shared between them. More recent solutions try to exploit the capabilities of neural networks to share information, either via their hidden states or via combining task-specific loss functions, thus capturing the interactions that exist between the two tasks.

### 2.1. Solving the Two Tasks Separately

Since it is modelled as a text classification task, intent detection has been traditionally tackled using various classifiers, such as Support Vector Machines (SVMs) [5], Naive Bayes and Logistic Regression [6], or boosting methods [7], generally on bag-of-words representations and domain-engineered features, or—more recently—to word embedding representations. SVM models are still currently present in commercial tools such as RASA NLU [4]. Specific deep learning architectures have also been explored for intent detection alone—see [8] for examples on the use of Convolutional Neural Networks (CNNs) or Recurrent Neural Networks (RNNs), respectively.

As a sequence tagging task, slot filling has been traditionally addressed via two approaches—either map it to a classification task with global features, or use some kind of sequence model—such as Conditional Random Fields (CRFs) or RNNs [9].

### 2.2. Joint Approaches

More recent techniques attempt to exploit the interactions and information sharing between the two tasks, by learning a joint classification and sequence labelling model. Various deep architectures are explored in this context—several which optimize some linear combination between the two different cost functions—one for the intent detection and one for slot filling—jointly, but there are even attempts to keep the two cost functions separate, and thus share only the latent representations, but updating them independently during training.

In [10], the authors explore using a CNN architecture to learn features for slot filling, and use a CRF-like model on top of the latent CNN representations to generate slot predictions. The same latent CNN representations are summed up and fed to a separate Softmax layer for intent detection. The intuition behind the idea is that the system is able to handle both tasks simultaneously due to the global nature of the sequence labeling component. The authors of [9] explore using several different architectures for learning joint latent representations for the two tasks. Both word and character-level embeddings are employed in all architectures (the latter being trained either via a CNN or LSTM), and all employ a baseline CNN model to learn the sentence representation. Then, the authors explore different hierarchical models, by adding layers to this baseline model—either MLP-based to generate intent and slot predictions directly, or another CNN followed by either MLP or CRF layers for either of the two tasks, or an RNN (BiLSTM or BiGRU) layer followed by either MLP or CRF as output layers.

Attention mechanisms on top of a BiLSTM for joint intent detection and slot filling are explored in [11]. The paper first explores how explicit slot alignment information in the decoder can boost the performance of a joint encoder–decoder architecture (the encoder being shared among the two tasks), and then shows that further adding attention to the decoder is beneficial from a performance point of view. Taking the idea further, the authors propose using a single BiLSTM architecture with attention (instead of an encoder–decoder model): slot dependency is captured by the forward LSTM, and attention is added to the concatenated hidden state representation which is used to predict both intent and slot labels. This model is more computationally efficient, and achieves the same performance as the attention encoder–decoder with explicit alignment information.

A bidirectional GRU with a combined loss function is explored in [12] for both tasks. The solution employs a max-pooling layer to generate the global learned representation for predicting the intent label. The loss function is a linear combination of the cross-entropy (for intent detection) and structured margin loss (for slot filling). Word context windows composed of concatenated word embedding vectors are used as input, together with specific named entity context window embeddings. The named entity embeddings are learned during the training process.

Instead of combining the loss functions of the two different tasks, [13] proposes combining the hidden states of two BiLSTMs which are trained asynchronously, each with its task-specific loss function. The authors argue that asynchronous training is beneficial to overcoming the structural limitations of using a single architecture for both tasks, while maintaining the information shared between the two tasks in the models learned. The paper proposes two architectural variations—with and without a separate decoder for the output, the former achieving better performance during the evaluations.

A joint, “recurence-less” framework that exploits the benefits of contextual word embeddings is introduced in [14]. The framework proposed is multilingual and employs a classifier and a sequence labeler on top of pre-trained BERT embeddings. The model uses separate output layers for the two tasks, but both consist of a single linear layer. The joint loss function is a parameterized linear combination of the intent level cross-entropy and the average token-level (i.e., slot) cross-entropy.

The Capsule-NLU architecture proposed in [15] attempts to exploit the hierarchical relationships between the words, slots and intent of an utterance: the slots are inferred from word features and the overall intent of the sentence is determined from the learned slots representation. The model is built using three types of capsules:WordCaps—used to learn word representations that take into account the context of the sentence. This is done using a bi-directional LSTM.SlotCaps—used for determining the slot type of each word. Using a Dynamic routing-by-agreement algorithm, each word representation given as output by the WordCaps is routed to those SlotCaps with which it has a strong agreement.IntentCaps—perform intent detection. The same Dynamic routing-by-agreement algorithm is used here as well, this time between the SlotCaps and IntentCaps.

Having the prediction for the intent, the authors propose a re-routing schema in which high-level features (the predicted intent) are used to improve the predictions of lower-level features (the slots). The authors show that through joint training, the performance of the model is higher than through separate or sequential training.

Our approach draws inspiration from [15] in considering the interaction between the two tasks in a joint manner, by using a capsule network architecture similar to the one proposed there. The differences consist in including a fourth capsule type introduced in [16], which uses a self-attention mechanism to extract sentence-level semantic information, which is then leveraged during intent detection.

## 3. Description of the Home Assistant Scenario

The home assistant scenario was initially introduced in [3], and consists of multiple datasets that include several data and learning challenges that are likely to appear in such an application domain, considering the Romanian language. To the best of our knowledge, this is the first and only dataset for intent detection and slot filling for the Romanian language to date. It consists of 14 intents, grouped into three categories: light, temperature and media control commands. Figure 2 illustrates a taxonomy of the intents, categorized by the super-class they belong to. The diagram is also color-coded to illustrate what slots each intent can have. An intent with multiple colors can be associated with any of the corresponding slot types. Table A1 in Appendix A contains this information in a tabular form, along with the Romanian translation for each intent name.

In this work we introduce a minor modification to the original dataset, by splitting the ChangeVolume intent into two opposing intents—IncreaseVolume and DecreaseVolume—to maintain a consistent approach to similar intents from other superclasses (e.g., IncreaseLightIntensity and DecreaseLightIntensity).

### 3.1. Data and Learning Challenges

Human language can be very nuanced and sometimes even ambiguous. Increased language variability is particular to the Romanian language, where there are many alternative words and word combinations that convey the same ideas, and often it is not just the individual words that add value and meaning to a sentence, but also the relations between them. Therefore, the ability to generalize from as many different sentence forms as possible, to look behind the ambiguity and to figure out never-before-seen words from context is one of the major challenges an NLU model is faced with. Having a model that is very good at generalizing to unseen language constructs is also important from the learning perspective, since collecting labeled data is, in general, a very time-consuming process, so we want to be able to use as little data as possible, while still obtaining a good performance. Additionally, we consider two important learning challenges, which occur in many real-world scenarios, and also translate to the home assistant scenario (to be detailed shortly).

Consequently, three major challenges are considered in generating the data used to train our intent detection and slot filling models:**Synonymy**, which refers to one meaning (intent) of the sentence being expressed by different language constructs. For instance, the home assistant should know that *“Turn on the lights in the kitchen”* and *“Start the lights in the kitchen”* have the same intent, knowing that the verbs *“turn on”* and *“start”* are synonyms in this context. This language challenge may translate to a covariate shift learning challenge [17], if the two constructs appear in the training/testing sets, respectively. Another situation can arise here when the same set of words can be used in a slightly different context, thus changing the intent of the sentence. For instance, *“Turn on the lights”* and *“Turn on the TV”*—the structure of the sentence is identical, the only difference lies in the object of the sentence. This language phenomenon also translates to a learning problem: class overlap problem [18].**Missing slot values**, dealing with slot values that haven’t been encountered during training, or sentences (intents) that are expected to have specific slots, but they are missing. For instance, *“Seteaza temperatura”* (*“Set the temperature”*) is missing an important piece of information indicating the value to which the temperature should be set, as well as the room in which to set it. However, the NLU model should not be confused by the missing slots and should correctly identify the intent and that there are no slots to be classified. Another example would be *“Play a song by The Rolling Stones”*—if this is the first time the user requests a song by this band, the home assistant should still be able to fulfill the command correctly. This language challenge maps to an incomplete data learning problem.**Class imbalance** [19], which in the context of intent detection and slot filling may appear with the emergence of new intents. As more and more parts of the house become automated, the NLU system has to be prepared for detecting new intents, new slot types and/or values with as little manual intervention as possible.

Another important aspect of the Romanian language is the presence of diacritics. Some words can be written with or without diacritics, which might affect their meaning in the context they appear in. The following example is not taken directly from our datasets, but it perfectly illustrates the ambiguity that can be introduced when the same word is written with and without diacritics:*“Am cumpărat niște pături de la magazin”*—*“I bought some blankets from the store.”**“Am cumpărat niște paturi de la magazin”*—*“I bought some beds from the store.”*

The only difference between these two sentences is the letter ă in the word pături/paturi, but this small difference is capable of changing the meaning of the sentence. So, by introducing diacritics in the data, we increase the problem complexity.

### 3.2. Scenarios and Data Description

Considering the data complexities in Section 3.1, multiple dataset instances (which we will also refer to as scenarios) have been generated incrementally: Scenario 1, which contains the first challenge, Scenario 2—containing the first two challenges and Scenario 3—containing all three challenges. A baseline dataset, representing the default scenario—Scenario 0, with no data complexities added to it—has also been considered, to allow for a “best case” performance assessment.

**Scenario 0—baseline**: this is the default dataset, in which the training and testing samples are drawn from the same uniform data distribution and contain the same vocabulary. This scenario is used to evaluate the performance of the NLU system in the general, ideal case without purposefully introducing challenging situations. It provides a baseline with which to compare the performance of the other scenarios.**Scenario 1**: this scenario deals with the primary challenge of using synonyms when expressing the same intent. The dataset, in this case, was generated in such a way that in the train set there are some formulations for expressing an intent, while in the test set there are different but synonymous ways of conveying the same intent. Some examples from both the train and the test set can be found in Table A2.**Scenario 2**: on top of the synonymy challenge, this scenario also contains the missing slots challenge. Here, in the training set the slot values are drawn from one set of values (for instance the Room slot can take one of four values: bucatarie, baie, curte, sufragerie—kitchen, bathroom, yard, living room), while in the test set there is a totally different set of values for that same slot (camara, dormitor, pivnita, camera de zi—pantry, bedroom, cellar, living room). Additionally, the test set includes sentences from which the slots are missing altogether. This is done in the case of intents which normally expect some types of slots and for which the sentences still make sense if no slots are present. Some examples for this scenario can be also be found in Table A2.**Scenario 3**: in this scenario, we took turns in imbalancing each class of intents (light, temperature and media), which resulted in three sub-scenarios (**3.1**, **3.2**, **3.3**), each consisting of different training and testing datasets. The imbalance is only present in the training set, with the number of samples from under-represented class being decreased by a factor of 10, while the test set keeps the same distribution as in the previous scenarios. For the exact quantities of intents in each scenarios, consult Section 5.1.

In order to save time and eliminate human error when manually labelling data, we decided to use a NLU dataset generation tool called Chatito [20]. This method has the added benefit of flexibility, allowing us to fine-tune the distribution of the datasets and engineer the challenges in each scenario.

## 4. Proposed Model Architecture

Similarly to how Convolutional Neural Networks learn to extract low-level features which are combined to form higher-level features as the information is propagated deeper into the network, Capsule Neural Networks [2] model the hierarchical relationships between low-level and high-level features. The key difference is that in a Capsule Neural Network, the relative pose information between parts of the object we want to detect is also preserved. The hierachical modelling capabilities of a Capsule Neural Network lend themselves to natural language applications, where lower-level word features are combined to obtain sentence-level features.

We propose a customized Capsule Neural Network architecture, SemCapsNet, which performs intent detection and slot filling jointly. The model is inspired by Capsule-NLU [15] and IntentCapsNet [16]. The basis for our implementation was the Capsule-NLU model. Because it uses only the slot information as an input for determining the intent of the sentence and because in our dataset we have multiple intents that contain the same slot types (e.g.,  IncreaseLightIntensity and IncreaseTemperature), we decided to augment the Capsule-NLU architecture with a type of capsule used in IntentCapsNet: the SemanticCaps, which extracts semantic features from the utterance. These new SemanticCaps provide another type of input for determining the intent of the sentence and should help discriminate between intents that have the same types of slots.

The architecture consists of four types of capsules:WordCaps [15] that learn word representations based on their context in the sentence.SlotCaps [15] that are used to classify each word into a slot type, as well as to learn slot representations from all the words classified to each slot type.IntentCaps [15] that determine the intent of the user utterance based on the extracted semantic features, as well as the slot representations.SemanticCaps [16] that extract semantic features from the input sentence.

The first three capsule types (WordCaps, SlotCaps, IntentCaps) are first proposed in [15], while the SemanticCaps are added from [16]. We introduce the SemanticCaps in the architecture at the same level as the SlotCaps, in order to provide additional input for determining the intent of the utterance. Figure 3 illustrates how the capsules are interconnected in SemCapsNet.

### 4.1. Input

The training set consists of labeled examples of the form <x,y,z>, where *x* = input sentence, *y* = intent label and *z* = sequence of slot labels.

Each utterance can be viewed as a sequence $x=(w_{1},\;w_{2},\;\ldots ,\;w_{T})$, where *T* is the number of words in the utterance and each word in the sentence is represented by a word vector of size $D_{W}$. The intent label *y* belongs to a set of *K* intents, $y\in \{y_{1},\;y_{2},\;\ldots ,\;y_{K}\}$, while each slot $z_{t}$ in the slot label sequence $z=(z_{1},\;z_{2},\;\ldots ,\;z_{T})$ is from a set of *L* slots, $z_{t}\in \left\{z_{t}^{1},\;z_{t}^{2},\;\ldots ,\;z_{t}^{L}\right\}$.

### 4.2. WordCaps

The purpose of the WordCaps [15] is to take the input sentence, whose words are represented as word vectors, and to learn word representations that take into account the context of the sentence they are part of. This is done using a Bidirectional Recurrent Neural Network (BiRNN) with LSTM cells:(1)$$\begin{array}{l} {\vec{h}}_{t}=LSTM_{fw}\left(w_{t},{\vec{h}}_{t-1}\right)\\ {\overset{\leftarrow }{h}}_{t}=LSTM_{bw}\left(w_{t},{\overset{\leftarrow }{h}}_{t+1}\right). \end{array}$$

The forward and backward hidden states ${\vec{h}}_{t}$ and ${\overset{\leftarrow }{h}}_{t}$ are concatenated into one hidden state $h_{t}$ per word. Thus, we obtain a hidden state matrix $H=\left(h_{1},\;h_{2},\;\ldots ,\;h_{T}\right)\in R^{T\times 2D_{H}}$ where $D_{H}$ is the number of hidden units in each LSTM cell.

### 4.3. SemanticCaps

Unlike the WordCaps, which extract context-aware representations on a word-level, the SemanticCaps [16] obtain relevant semantic features from the overall sentence. This is done using a self-attention mechanism with multiple attention heads.

The attention matrix *A* is computed from the hidden state matrix *H* as follows:(2)$$A=softmax(W_{s2}tanh\left(W_{s1}H^{T}\right)),$$
where $W_{s1}\in R^{D_{A}\times 2D_{H}}$ and $W_{s2}\in R^{R\times D_{A}}$ are weight matrices, $D_{A}$ is the number of self-attention hidden units and *R* is the number of self-attention heads. To actually extract the semantic features, the attention matrix *A* is multiplied with the hidden state matrix *H*:(3)M=AH,
where $M=\left(m_{1},\;m_{2},\;\ldots ,\;m_{R}\right)\in R^{R\times 2D_{H}}$. There are *R* SemanticCaps in total, one for each extracted semantic feature.

### 4.4. SlotCaps

Now that we have a learned vector representation for each word in the sentence, we use SlotCaps [15] to determine the slot labels. SlotCaps use the WordCaps output to learn sentence-level vector representations for each slot type via Dynamic Routing by Agreement [2].

First, slot prediction vectors are computed between each pair of word and slot capsules:(4)$$p_{lt}=\sigma \left(W_{l}h_{t}^{T}+b_{l}\right),$$
where $p_{lt}$ is the prediction vector of the *t*-th word being labelled as having the *l*-th slot type, l∈{1, 2, ⋯, L}. $W_{l}\in R^{D_{L}\times 2D_{H}}$ and $b_{l}\in R^{D_{L}\times 1}$ are the weight and bias matrices for the *l*-th slot capsule and σ is the activation function (in our case we apply tanh). $D_{L}$ is the dimension of the slot prediction vector, $p_{lt}\in R^{D_{L}\times 1}$.

Then, Dynamic Routing by Agreement (Algorithm 1) is applied on the prediction vectors $p_{lt}$ to obtain the SlotCaps output vectors $v_{l}$:
*v_l_* = DynamicRouting(*p_lt_*, 1, *iter_slots_*)(5)
**Algorithm 1** Dynamic routing by agreement [2].1:**procedure**DynamicRouting($p_{ji},l,iter$)2: for all capsules *i* in layer *l* and *j* in layer l+1: $b_{ij}\leftarrow 0$3: **for**
iter iterations **do**4:  for all capsules *i* in layer *l*: $c_{i}\leftarrow softmax\left(b_{i}\right)$5:  for all capsules *j* in layer l+1: $s_{j}\leftarrow \sum_{i}c_{ij}p_{ji}$6:  for all capsules *j* in layer l+1: $v_{j}\leftarrow squash\left(s_{j}\right)$7:  for all capsules *i* in layer *l* and *j* in layer l+1: $b_{ij}\leftarrow b_{ij}+p_{ji}\cdot v_{j}$8: **end for**9: **return**
$v_{j}$10:**end procedure**

### 4.5. IntentCaps

The overall intent of the utterance is identified based on the predicted slots, as well as the extracted semantic features, so the output vectors of the SemanticCaps and SlotCaps are fed into the IntentCaps [15]. Since the original IntentCaps only receive the output of SlotCaps as input, we adapt their dimensions so as to take into account the SemanticCaps output as well. We concatenate the two output vector matrices $M=\left(m_{1},\;m_{2},\;\ldots ,\;m_{R}\right)\in R^{R\times 2D_{H}}$ and $V=\left(v_{1},\;v_{2},\;\ldots ,\;v_{L}\right)\in R^{L\times 2D_{H}}$ to obtain the layer 2 output vector matrix $N=\left(m_{1},\;m_{2},\;\ldots ,\;m_{R},\;v_{1},\;v_{2},\ldots ,\;v_{L}\right)=\left(n_{1},\;n_{2},\;\ldots ,\;n_{G}\right)\in R^{G\times 2D_{H}}$, where G=L+R and $D_{L}=2D_{H}$.

The IntentCaps take the output vectors from the capsules in the previous layer and try to establish the overall intent of the sentence. First, the intent prediction vectors are computed:(6)$$q_{kg}=\sigma \left(W_{k}n_{g}^{T}+b_{k}\right),$$
where k∈{1, 2, ⋯, K}, g∈{1, 2, ⋯, G}, $q_{kg}\in R^{D_{K}\times 1}$. $W_{k}\in R^{D_{K}\times 2D_{H}}$ and $b_{k}\in R^{D_{K}\times 1}$ are the weight and bias matrices for the *k*-th intent capsule and $D_{K}$ is the dimensionality of the IntentCaps prediction and output vectors.

To obtain the IntentCaps output vectors, the same dynamic routing algorithm as before is applied on the prediction vectors:
*u_k_* = DynamicRouting(*q_kg_*, 2, *iter_intents_*).(7)

The resulting output vectors $u_{k}\in R^{D_{K}\times 1}$ indicate the intent type of the utterance.

### 4.6. Rerouting

So far, the hierarchical modelling capabilities of capsule networks have been used so that low level features help in determining the higher-level features, i.e., the slot predictions help drive intent prediction. However, intents can also be used to further refine the slot prediction, since slots are tied to one or more intents—once we have a prediction for the intent, the slot filling step in the architecture can be revisited and those slots which fit with the predicted intent can be boosted.

To achieve this, the rerouting schema proposed in [15] is applied between the IntentCaps and the SlotCaps. The dynamic rerouting is achieved by changing line 7 in Algorithm 1 to the following:(8)$$b_{ij}\leftarrow b_{ij}+p_{ji}\cdot v_{j}+\beta \cdot p_{ji}^{T}W_{RR}{\hat{u}}_{z}^{T},$$
where ${\hat{u}}_{z}^{T}$ is the activation vector of the predicted intent, β is a rerouting coefficient and $W_{RR}$ is the rerouting weight matrix.

### 4.7. Loss Functions

For intent detsection, we use the max-margin loss [2]:(9)$$\begin{array}{c} L_{k}=T_{k}max{\left(0,m^{+}-∥u_{k}∥\right)}^{2}+\lambda \left(1-T_{k}\right)max{\left(0,∥u_{k}∥-m^{-}\right)}^{2}\\ T_{k}=\left\{\begin{array}{cc} 1, & \mathrm{true}\;\mathrm{intent}\;\mathrm{is}\;k\\ 0, & \mathrm{otherwise} \end{array}\right. \end{array}$$

To this, an extra regularization term for the self-attention mechanism is added, as in [16]:(10)$$P=\alpha {∥AA^{T}-I∥}^{2}$$
where α is a trade-off coefficient that helps with keeping each self-attention head attentive to different areas of the sentence.

Thus, the total intent detection loss $L_{intent}$ is:(11)$$L_{intent}=\sum_{k}L_{k}+\alpha {∥AA^{T}-I∥}^{2}.$$

We model slot filling as a multi-class classification task. Therefore, we employ the cross-entropy loss for this part of the learning problem:(12)$$L_{slot}=-\sum_{t}\sum_{l}z_{t}^{l}log\left({\hat{z}}_{t}^{l}\right).$$

The total loss is simply the sum of the two losses:(13)$$L_{total}=L_{intent}+L_{slot}.$$

## 5. Empirical Evaluations

### 5.1. Quantitative Description of the Datasets

The datasets generated according to the scenarios described in Section 3.1 contain 14 intents in total, each having 90 examples in the training set and 40 examples in the test set. This is true for the baseline scenario and Scenario 1. In Scenario 2, we considered only 12 of the 14 intents with the same train/test split of 90/40, as the 2 intents StopMusic/OpresteMuzica and StopTV/OpresteTV do not have any corresponding slots and can be omitted from this scenario. Lastly, in Scenarios 3.1, 3.2 and 3.3 the intents corresponding to the light, temperature and media intent super-classes respectively are underrepresented, with only nine examples in the train set and 40 examples in the test set. In each of these three sub-scenarios, the intents belonging to the other two super-classes have the same train/test split of 90/40, as in the previous scenarios. All the datasets employed in the evaluations and the code are available at: https://github.com/keg-tucn/home-assist-capsnet accessed on 10 December 2020.

### 5.2. Experimental Setup

The capsule network architecture is trained using 3-fold cross-validation and the Adam optimizer [21]. Table 1 contains the hyperparameter values used for training in all scenarios. $D_{H}$ is the number of hidden units in the BiRNN, $D_{L}$ and $D_{K}$ are the SlotCaps and IntentCaps output vector dimensions, respectively. $D_{A}$ is the number of self-attention hidden units, while *R* controls the number of self-attention heads. α is the self-attention regularization coefficient and β is the rerouting coefficient. Lastly, $iter_{intent}$ and $iter_{slot}$ indicate the number of iterations performed by the dynamic routing algorithm in the case of the IntentCaps and SlotCaps, respectively.

### 5.3. Quantitative Result Analysis

We perform experiments using multiple word vector models when training the capsule architecture. Initially, we started with a 300-dimensional fastText model (https://fasttext.cc/docs/en/crawl-vectors.html accessed on 10 December 2020). We then reduced the dimensionality of the word vectors to 100 and 50. The results of the three models are reported in Table 2. In the remainder of the paper, we denote a SemCapsNet architecture trained using the x-dimensional fastText word vectors by SCN-x. We measure the performance of intent detection and slot filling using the F1-score. Models trained using the Rasa NLU tool serve as a baseline against which we compare the capsule network results.

The model trained on the fastText 100-dimensional vectors (SCN-100) achieves overall better results, apart from intent detection in the baseline scenario, where the higher dimensional SCN-300 performed better. However, in both of these situations, the increase from the SCN-100 score is almost negligible, while for all the other scenarios the SCN-100 model leads to significantly higher performance. Slot filling in S1 and S3.3 reaches a slightly higher score when using the SCN-50 model. The third scenario has the lowest performance, which was expected since it is the most complex one.

It is worth noting that the first added complexity (synonymy) produces a dramatic deterioration of the intent detection performance (of over 40%), while the other complexities added on top of it introduce mild supplementary degradation (~4% for missing slots and ~6% on average for super-class imbalance).

Looking at the results achieved by the Rasa NLU models, we can see that the difference between using the 300-dimensional word vectors versus the 100-dimensional ones is not as drastic: ~2% at most in the baseline scenario, while the other scenarios show a difference of ~1% or less. Additionally, we notice that the Rasa models generally perform better when it comes to slot filling than SCN, with Rasa-300 reaching a significantly higher slot filling performance in Scenarios 1 and 3.3 (~11% and ~21%, respectively) and Rasa-100 doing better in Scenarios 0, 2 and 3.2 by a smaller margin (less than 2%). When it comes to intent detection, however, the SCN models perform better across the board, with differences starting from ~8% and going as high as ~19% in Scenarios 1 through 3.3.

### 5.4. Qualitative Result Analysis

We carry out a more detailed error analysis using the best performing model (SCN-100). We notice that many of the error types that were present in the Rasa NLU model [3] are also produced by the capsule network.

#### 5.4.1. Intent Detection

There are three main types of intent errors that we have observed, which we codify as ERR_OPP, ERR_ACT and ERR_CAT. The description of each error type along with some examples can be found in Table 3. Not all the errors can be classified in one of these categories, however, the majority of them can be, so these are the error types that we focus on in our analysis.

Figure 4 contains a visual representation of how the error types are distributed through the dataset. The intents taxonomy previously shown in Figure 2 is now color-coded by error types in order to illustrate which pairs or groups of intents are affected by each error type. Different shades of the same color are used to separate between groups of intents affected by the same error (e.g., ERR_ACT occurs between TurnOnLight and StartTV, as well as between StopMusic and StopTV).

Table 4 shows the distributions of the error types in each scenario, starting with Scenario 1. The values reported for each error type represent the percentage of errors that fall within that category, out of the total intent errors in the scenario.

Even in the baseline scenario, out of the 4 total intent errors, 1 is of type ERR_OPP (IncreaseVolume mixed up with DecreaseVolume) and 1 of type ERR_ACT (DecreaseVolume mixed up with DecreaseLightIntensity).

The confusion matrices in Figure 5 provide a visual representation of the frequency of these error types. In Figure 5a, we can see that in Scenario 1 the most frequently occurring confusions fall into one of the three error types. Comparing this with the confusion matrix of Scenario 3.1 (Figure 5c) we notice that the recurring confusions are still those of either type ERR_OPP, ERR_ACT or ERR_CAT. There are few more non-typical errors, especially in the case of intents belonging to the light super-class, but that is expected considering that in Scenario 3.1 that super-class is underrepresented.

To get a sense of how well the model is doing at keeping the predicted intent in the same super-class (light, temperature, media) as the true intent, we also plot super-class confusion matrices (Figure 5b,d). In Scenario 1, all of the media and temperature intents are predicted within the same super-class and even the majority of the light intents are kept in the same super-class, here most of the errors being of type ERR_OPP. The confusion and super-class confusion matrices for the other scenarios can be found in Figure A6 in Appendix A.

Looking more closely at the error type ERR_ACT, this occurs when two sentences belonging to different intents use the same verb, or synonyms of that verb, but refer to a different object (like TurnOnLight and StartTV). In such situations, the verb can be used correctly in both contexts and it appears that it has a higher influence on the final predicted intent than the other words in the sentence which might be more specific to the true intent. In the case of the previously mentioned intents (TurnOnLight and StartTV) the verb *“pornește”*/*“start”* was seen during training under the StartTV intent, while TurnOnLight was trained with the verb *“aprinde”*/*“turn on”*. At testing time however, “pornește” was used in the context of TurnOnLight and in around half of the incorrectly predicted intents, was predicted as StartTV (ERR_ACT), while the other half was predicted as its opposite intent TurnOffLight (ERR_OPP).

#### 5.4.2. Slot Filling

As far as slot filling is concerned, there are some recurring patterns of errors there as well. We have noticed that many of the slot filling errors are due to parts of the slot not being labeled correctly when a slot is formed of multiple words in the sentence. Either one or more words belonging to the slot are ignored, or the labels do not follow the IOB format. For instance *“douăzeci și șapte de grade”*/*“twenty seven degrees”* is labeled as I-grade I-grade … I-grade instead of B-grade I-grade … I-grade or *“Pink Floyd”* is labeled as B-artist B-artist instead of B-artist I-artist. This error type occurs most frequently with the degrees (grade) and artist slots. In the case of the artist slot, the error could appear due to the model never having seen during training an artist name formed of more than one word, so it doesn’t know about the I-artist label. However, in some situations, the artist name is only partly labeled, with some words being ignored completely, so the previous argument only partly justifies the errors that appear for this particular slot type.

Table 5 shows the distributions of slot filling errors in the super-classes, starting with Scenario 1. The values reported for each error type represent the percentage of errors that fall within that super-class, out of the total slot filling errors in the scenario.

We notice that the temperature super-class is the one that is most affected by slot filling errors in almost all scenarios. An exception to this is Scenario 3.3, in which the media super-class has a higher percentage of errors, but that is to be expected as the media intents are underrepresented in the scenario.

With this error distribution in mind, the sudden drop in slot filling performance exhibited by the SCN-100 model and reported in Table 2 for Scenarios 3.2 and 3.3 could be explained by the fact that the temperature and media super-classes are generally the ones encountering the most errors with slot filling, the problem further exacerbated by the added imbalance.

We have also discovered that, starting with Scenario 1, when the intent is incorrectly classified the slot prediction is also driven towards a slot type that corresponds with the predicted intent. An example would be the TurnOnLight intent: when it is classified as StartTV, the slot indicating the location is labeled as a channel slot, specific to StartTV (there are 13 examples classified as StartTV, out of which six have their slots labeled incorrectly). On the other hand, when TurnOnLight is classified as TurnOffLight, there are fewer slot filling errors (out of 11 examples classified as TurnOffLight, two have their slots labeled incorrectly using a channel type slot). This could be in part due to the rerouting mechanism, however, we have noticed this pattern when training the model without rerouting as well.

#### 5.4.3. Confidence Levels

The confidence levels of the predictions can also offer some insight into how the model is operating. We consider the confidence level of an intent *k* to be the norm of the vector $u_{k}$, output by the *k*-th intent capsule. We use these values to plot confidence matrices that show some confidence score with which the intent *x* is predicted as *y*.

We compute the confidence score ${\mathrm{CS}}_{xy}$ as the average confidence of *x* predicted as *y*:(14)$${\mathrm{CS}}_{xy}=\frac{1}{m_{xy}}\sum_{i}C_{xy}^{i}$$
where $C_{xy}^{i}$ is the confidence of the *i*-th prediction of *x* as *y* and $m_{xy}$ is the number of examples of intent *x* that are predicted as *y*.

The confidence matrix for Scenario 1 can be seen in Figure 6.

The main diagonal of the matrix corresponds to the average confidences of correct predictions, which are higher than 50% for almost all intents—the exceptions are DecreaseTemperature which is always predicted as SetTemperature (ERR_CAT) and StartTV which has an only slightly lower confidence at 48.96%.

Looking at the values off the main diagonal, we can notice that for most incorrectly predicted intents the highest score is obtained when the true and predicted intent are from the same super-class. Moreover, the highest confidence was obtained for errors that can be classified in one of the three error types presented before in Table 3:DecreaseVolume and IncreaseVolume—85.98%, ERR_OPPDecreaseLightIntensity and IncreaseLightIntensity—84.38%, ERR_OPPIncreaseLightIntensity and TurnOnLight—82.21%, ERR_CATIncreaseTemperature and SetTemperature—77.67%, ERR_CATDecreaseTemperature and SetTemperature—75.65%, ERR_CATStopTV and StopMusic—63.98%, ERR_ACTTurnOffLight and TurnOnLight—62.60%, ERR_OPP

It is important to keep in mind that this matrix shows only the average confidence, so a high value could correspond to very few errors made with a high confidence (for instance, in the case of IncreaseLightIntensity predicted as TurnOnLight). In order to get the full picture, this chart is best interpreted together with the confusion matrix, since it can give additional information regarding the frequency of a certain prediction. With this approach, we can see that whenever the model repeats the same prediction for an intent, it does so with a high confidence, which is to be expected. This happens both for correct predictions (e.g., IncreaseLightIntensity, SetTemperature and PlayMusic), as well as for incorrect ones (e.g., IncreaseTemperature and DecreaseTemperature predicted as SetTemperature, and DecreaseVolume predicted as IncreaseVolume).

The confidence matrices for Scenarios 0, 2, 3.1, 3.2 and 3.3 can be found in Appendix A, Figure A1, Figure A2, Figure A3, Figure A4 and Figure A5, respectively.

### 5.5. Learning Curves

During the development process, an important question that emerged was related to the quantity of data that should be generated for training a model that shapes the domain well, without under- or over-fitting. To answer this, we analyze the learning curves indicating how the performance of the model changes when varying the size of the training set. We build separate models for different numbers of training examples per class (TEC) and evaluate the loss on the train and validation sets. For the train set, we set up an initial dataset of TECInit examples and with each consecutive model built we incorporate an additional number of training samples per class (Inc), up to a maximal number (TECMax). In our experiments we consider: TECInit = 5, Inc = 5, TECMax = 60. The remaining 30 examples per intent (up to 90) are added to the validation set (VEC = 30). At the end of the training, we estimate the loss function on both the train and validation sets and plot the values on the same diagram (Figure 7).

We notice that both the train and validation losses are decreasing, with the validation loss reaching almost 0 at around 35 examples in Scenario 0 and at around 40 examples in Scenario 1. There are some spikes in the validation loss that could be attributed to only one model per dataset size being trained, which makes the model sensitive to variations in weight initialization from one training run to the other. However, once the optimal TEC value is reached, the loss settles around 0.

Thus, we could build a reliable model on a smaller dataset, with an optimal size of TECOpt = 35 for Scenario 0 and TECOpt = 40 for Scenario 1. This is most likely due to the nature of the dataset, which contains mostly short sentences, with a limited vocabulary.

### 5.6. Ablation Study

In order to explore the impact of adding self-attention via the SemanticCaps, as well as the benefit of rerouting, we perform an ablation study. We train separate models by removing in turn from the architecture the SemanticCaps, then the rerouting step, and finally both the SemanticCaps and the rerouting. The results of the study are reported in Table 6.

SCN-100 w/o att is the model without self-attention, which is equivalent to the CapsuleNLU model from [15]. In this model, the IntentCaps receive as input only the output vectors $v_{l}$ of the SlotCaps, such that $N=\left(v_{1},\;v_{2},\;\ldots ,\;v_{L}\right)\in R^{L\times 2D_{H}}$, and the loss function does not contain the regularization term *P*:(15)$$L_{intent}=\sum_{k}L_{k}.$$

SCN-100 w/o rr contains the SemanticCaps but does not perform rerouting. Finally, SCN-100 w/o att and rr does not contain either SemanticCaps or the rerouting mechanism.

It is interesting to notice that the SCN-100 w/o rr model, with self-attention but without rerouting, outperforms the regular model in slot filling in almost all scenarios (apart from S0 by a very small margin and S3.1). However, when it comes to intent detection, the regular model (with attention and rerouting) is the best performing one in all scenarios apart from S1. The idea behind rerouting was to revisit the slot filling step and refine the predictions there, once the intent information is obtained. We observe that for our use case, rerouting actually has the opposite effect and is in fact helping the intent detection process. Initially, a potential justification for this was based on the observation that when the intent is misclassified, it causes the slots to be predicted so that their type fits with the predicted intent (see Section 5.4.2). This could have been attributed to rerouting, as an incorrect intent prediction could influence the model to reconsider the slots during rerouting. However, we have observed that this is not the case, as the same behavior occurs even when the model does not use rerouting at all.

#### 5.6.1. Attention Values

To investigate the self-attention mechanism and which parts of the sentence the model considers to be important, we visualize the attention scores that each attention head produces on every word of an utterance. These values are found in the attention matrix $A\in R^{R\times T}$, in which each of the *T* words in the sentence has some attention scores assigned to it, one for each of the *R* attention heads. The larger the score of a word, the more the attention head is focused on it. Figure 8 follows similar examples from the TurnOnLight intent, starting from Scenario 0 and going all the way to Scenario 3.3. We can observe how the attention shifts on some parts of the sentence, depending on the complexity that was introduced with each scenario.

Our expectation was that the attention would be focused on words or groups of words specific to each intent or on parts of speech that are more likely to indicate the correct intent (e.g., verbs or nouns). In Scenario 0 (Figure 8a), this expectation is confirmed as the attention heads are mostly focused on the verb in the sentence (*“pornești”/“turn on”*) and on the slot (*“curte”/“yard”*). As we add complexities in each subsequent scenario, the attention shifts on other words: in Scenario 1 (Figure 8b), every attention head moves its focus on the slots, with very high importance placed on the preposition preceding the slot (*“in”*). This is most likely due to the fact that the focus shifts on words that the model has already seen in the context, such as the slot values, as opposed to the rest of the sentence which now contains new formulations for that intent. We believe that the verb becomes ignored by the self-attention heads because the vocabulary of our dataset is quite limited, without a lot of variation in formulation within an intent or even within similar intents (mostly those involved in the three error types). Because of this, similarities between word vectors from the training set and new word vectors from the testing set are considered less important than words that have already been encountered multiple times throughout training. In Scenario 2 (Figure 8c), the attention is more spread out, due to both the new formulations and the new slot values that appear in the test set. Scenario 3 and its subscenarios are more difficult to interpret, as Scenarios 3.1 (imbalanced TurnOnLight) and 3.3 are very focused on the slots, while Scenario 3.2 is more similar to Scenario 2.

## 6. Discussion

The results of the evaluations performed have indicated that language and learning complexities—which are expected to appear in such application scenarios—can greatly impact the performance of intent detection and slot filling. Even if the proposed SemCapsNet architectures perform generally better than Rasa NLU models, they are also affected by these complexities, and the qualitative analysis of the results have indicated several potential issues. Language variability seems to have the largest effect on the performance of the learning models. The drop in performance from the baseline to Scenario 1 is the largest and the qualitative analysis suggests that the model tends to memorize specific words rather than learn the meaning of the sentence. We believe this to be rooted in the fact that we use word embeddings to represent words, which are good at expressing semantic similarity, but fail to capture certain relationships such as synonymy and antonymy (as shown in [3]). In order to mitigate this issue, an improvement could be to intervene in the word vector model and adjust the embeddings. As seen in Section 5.6.1 the attention is not placed on the verbs, but rather on parts of the sentence the model has seen before. By pushing synonyms closer together we hypothesize that some of the attention would be directed to the verbs in the sentence, thus managing to better learn the meaning of the sentence. Furthermore, as seen in Section 5.4.1, ERR_OPP is a frequently occurring error type, which could relate to the fact that antonyms are not well represented by the word vector model. Pulling the antonym word vectors farther away from each other could result in an improvement of the model when it comes to detecting opposing intents.

We also monitored the training and validation losses during training, and observed that the model is not overfitting (Figure 9). The validation loss is decreasing and is slightly lower than the training loss. This phenomenon could be the result of having similar samples in the validation and train datasets, paired with the fact that the validation loss is evaluated at the end of each epoch, while the training loss is evaluated more frequently, after each batch. It is also worth noting that the validation loss exhibits small periodic spikes after finishing an epoch of training. This could be the result of not shuffling the validation set after each epoch.

## 7. Conclusions

In this work, we explore a customized capsule neural network model, trained to classify intents and their slots in a Romanian home assistant scenario. We perform a series of experiments on a previously defined dataset that incorporates increasing degrees of difficulty, in order to test the resilience of the model in the face of data complexities and learning challenges. Upon performing the evaluations, we found that the model is most affected by language variability between the train and test sets. Moreover, we show that the addition of self-attention capsules improves the performance of the model, obtaining significantly better intent detection results than the models trained with Rasa NLU (as shown in Table 2).

## Figures and Tables

**Figure 1 sensors-21-01230-f001:**
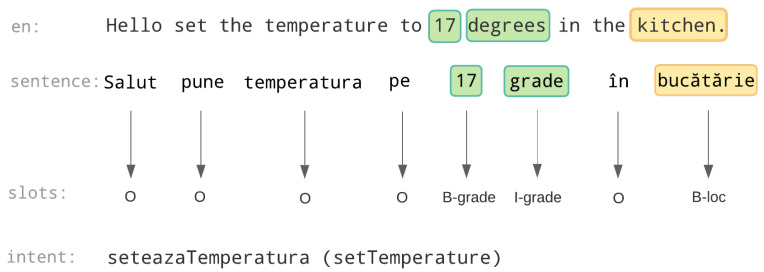
Intent detection and slot filling. For the slot filling labels, we use the IOB labelling format [1]. The “B-” prefix indicates that the word is at the beginning of a slot, the “I-” prefix indicates that the word is on the inside or at the end of a slot (occurs when slots contain at least 2 words). The “O” label is assigned to words not belonging to any slot.

**Figure 2 sensors-21-01230-f002:**
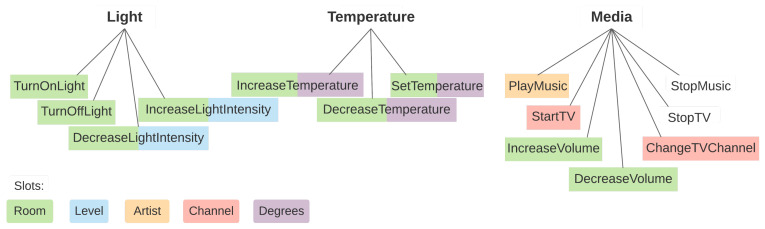
Intents taxonomy, color coded by slots.

**Figure 3 sensors-21-01230-f003:**
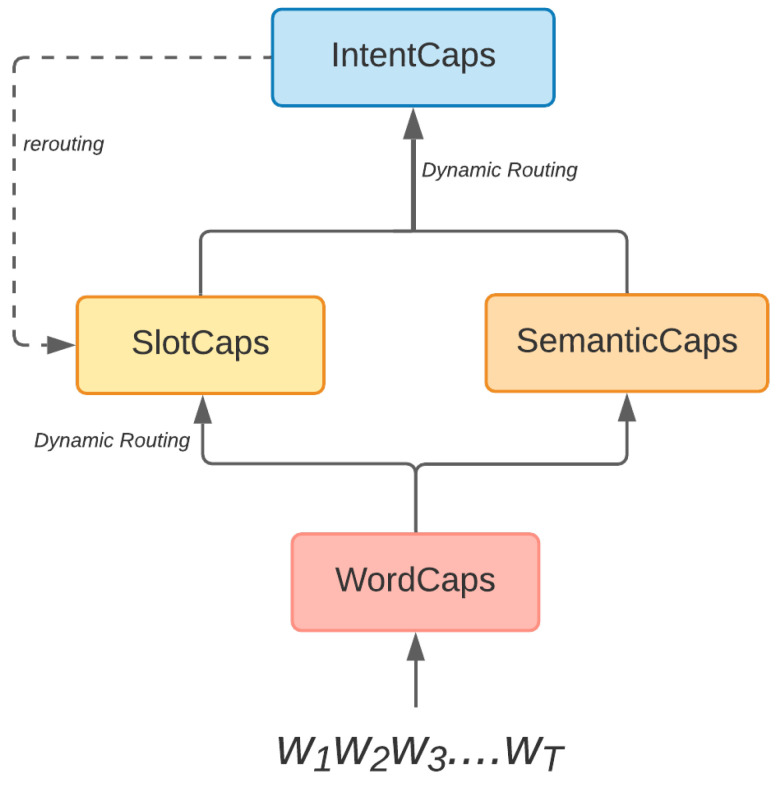
SemCapsNet architecture.

**Figure 4 sensors-21-01230-f004:**
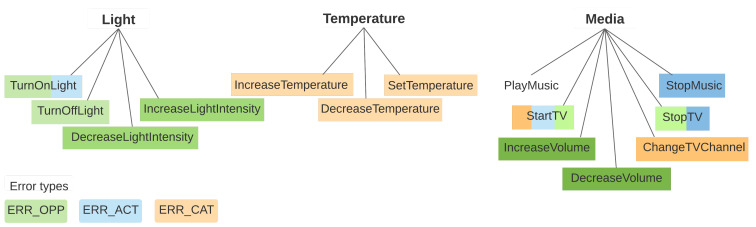
Intents taxonomy, color coded by error types.

**Figure 5 sensors-21-01230-f005:**
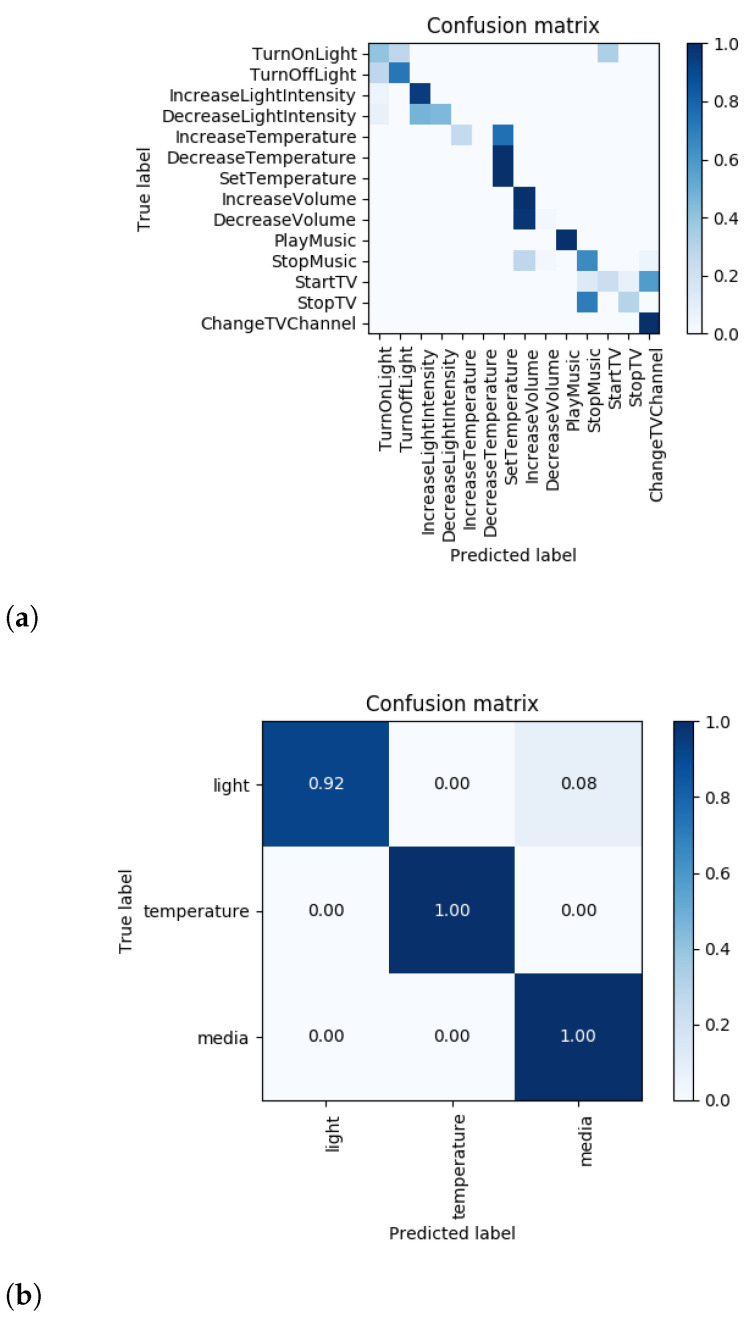
Intent confusion matrices (**a**) Scenario 1 (**b**) Scenario 1 Super-class confusion (**c**) Scenario 3.1 (**d**) Scenario 3.1 Super-class confusion.

**Figure 6 sensors-21-01230-f006:**
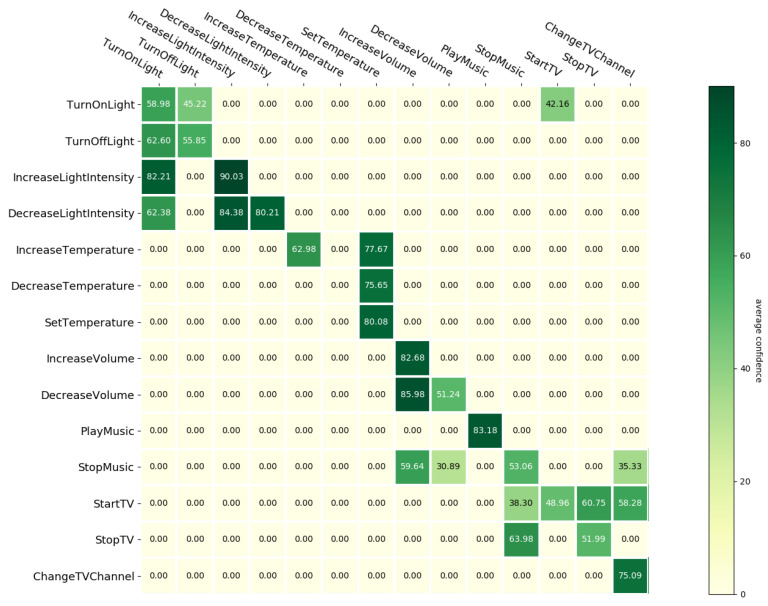
Confidence matrix for Scenario 1.

**Figure 7 sensors-21-01230-f007:**
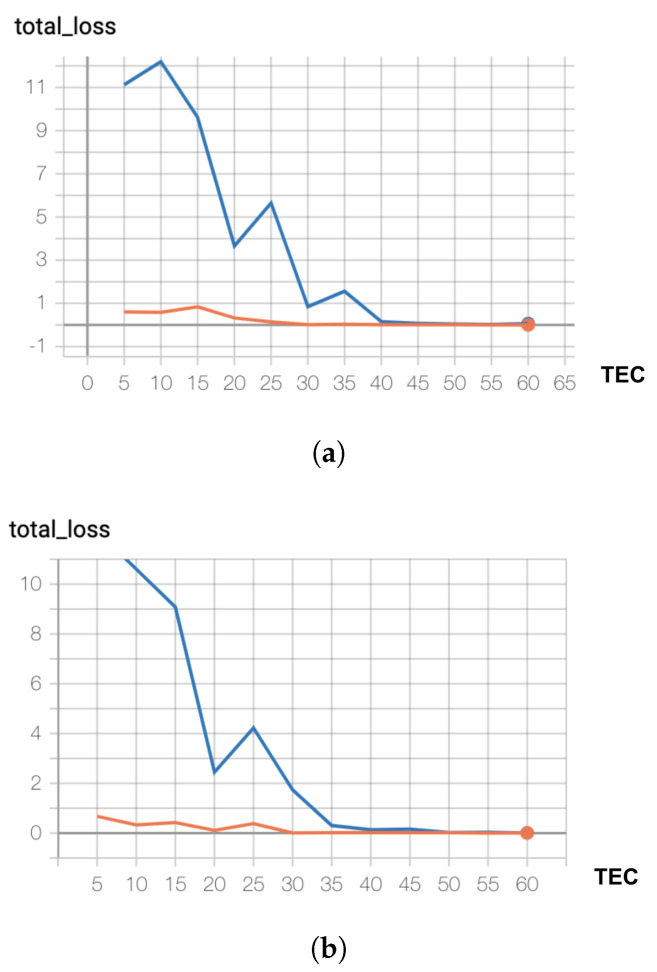
Learning curves: orange—train set loss, blue—validation set loss (**a**) Scenario 0 (**b**) Scenario 1.

**Figure 8 sensors-21-01230-f008:**
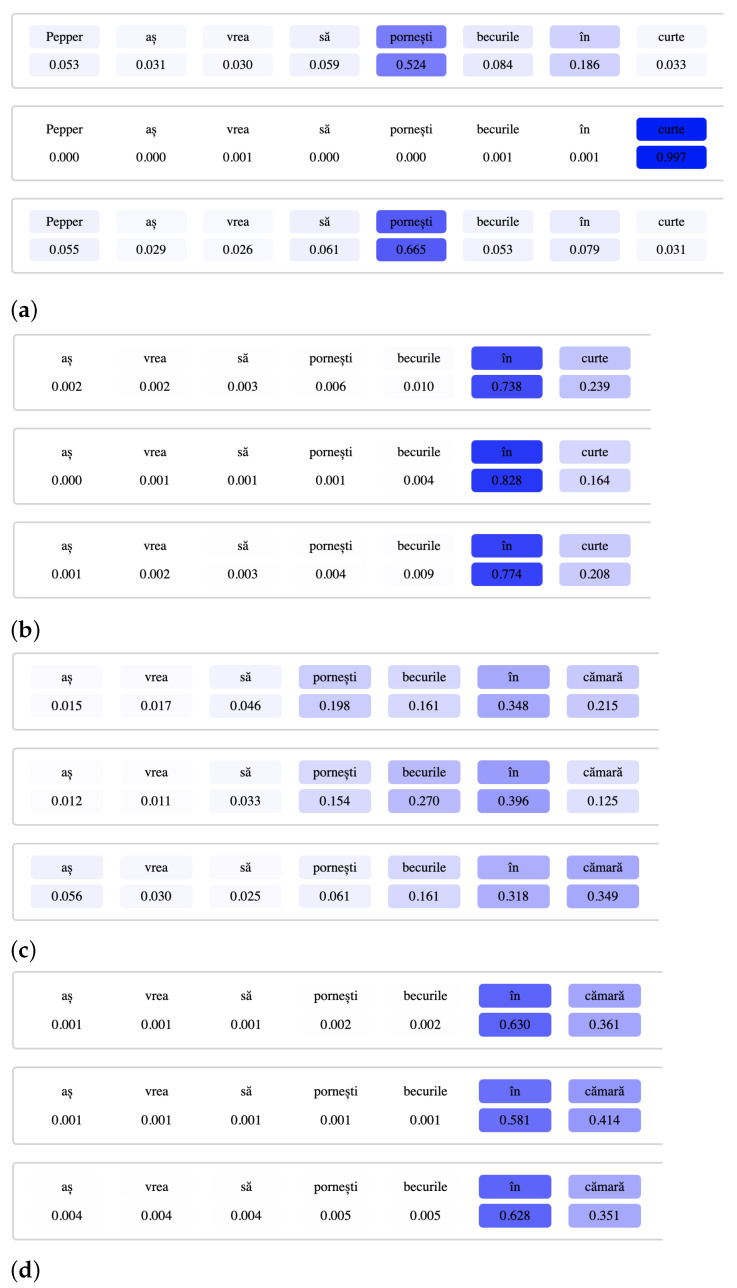
Attention scores on *“aș vrea să pornești becurile în …”*/*“I’d like you to turn on the lights in …”* (**a**) Scenario 0 (**b**) Scenario 1 (**c**) Scenario 2 (**d**) Scenario 3.1 (**e**) Scenario 3.2 (**f**) Scenario 3.3.

**Figure 9 sensors-21-01230-f009:**
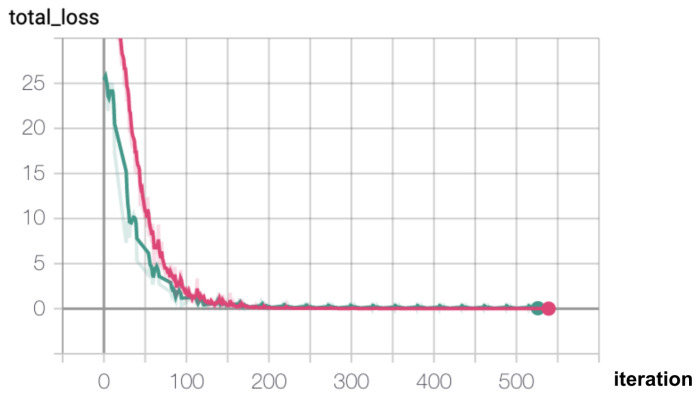
Training (magenta) and validation (green) losses.

**Table 1 sensors-21-01230-t001:** Hyperparameter values.

$D_{H}$	$D_{L}$	$D_{K}$	$D_{A}$	*R*	α	β	${\mathrm{iter}}_{\mathrm{intent}}$	${\mathrm{iter}}_{\mathrm{slot}}$
16	32	8	10	3	0.0001	0.5	3	2

**Table 2 sensors-21-01230-t002:** F1-scores obtained by the SemCapsNet and Rasa models using different word embedding models (best results are in boldface).

Model	Baseline		S 1		S 2		S 3.1		S 3.2		S 3.3	
	Intent	Slot	Intent	Slot	Intent	Slot	Intent	Slot	Intent	Slot	Intent	Slot
SCN-300	**99.64**	99.24	37.32	74.21	34.17	45.16	38.04	43.61	29.82	36.51	30.41	51.71
SCN-100	99.29	**99.69**	**56.96**	77.27	**52.50**	**64.32**	**48.57**	**70.92**	**46.07**	**50.04**	**45.26**	52.19
SCN-50	99.29	98.85	52.68	**77.73**	42.08	60.27	33.21	54.77	40.54	43.49	26.83	**52.85**
Rasa-300	98.67	**99.75**	48.39	**88.71**	43.07	64.36	31.07	64.99	35.16	51.49	25.74	**74.08**
Rasa-100	96.70	**99.75**	47.37	88.49	42.01	**64.52**	31.78	64.78	36.46	**51.90**	24.19	73.99

**Table 3 sensors-21-01230-t003:** Intent error types.

Error	Description	Example
ERR_OPP	Confusion between intents that are opposites/antonyms	TurnOnLight and TurnOffLightStartTV and StopTV
ERR_ACT	Same action, but on a different object	TurnOnLight and StartTV
ERR_CAT	Same class of intents, same object of an action but the actions themselves are not quite opposites	IncreaseTemperature and SetTemperatureDecreaseTemperature and SetTemperatureStartTV and ChangeTVChannel

**Table 4 sensors-21-01230-t004:** Intent detection—error type distribution (%).

Scenario	ERR_OPP	ERR_ACT	ERR_CAT	Other
S1	34.44	17.01	38.59	9.96
S2	46.93	6.58	32.89	13.60
S3.1	32.29	17.71	28.82	21.18
S3.2	38.08	14.57	27.15	20.20
S3.3	36.60	14.71	26.80	21.90

**Table 5 sensors-21-01230-t005:** Slot filling—error distribution in super-classes.

Scenario	Light	Temperature	Media
S1	11.33	52.67	36.00
S2	29.90	39.71	30.39
S3.1	32.54	41.15	26.32
S3.2	31.46	45.54	23.00
S3.3	17.92	38.33	43.75

**Table 6 sensors-21-01230-t006:** Ablation study results.

Model	Baseline		S 1		S 2		S 3.1		S 3.2		S 3.3	
	Intent	Slot	Intent	Slot	Intent	Slot	Intent	Slot	Intent	Slot	Intent	Slot
SCN-100	**99.29**	**99.69**	56.96	77.27	**52.50**	64.32	**48.57**	**70.92**	**46.07**	50.04	**45.26**	52.19
SCN-100 w/o att ^1^	98.57	99.31	55.18	70.52	40.62	61.05	35.18	57.80	43.57	51.11	25.76	51.95
SCN-100 w/o rr	99.29	99.31	**61.96**	**81.74**	45.00	**68.04**	46.96	65.30	43.39	**52.18**	41.68	**52.64**
SCN-100 w/o att and rr	97.14	93.03	51.07	74.96	29.58	60.37	25.89	44.22	39.29	36.22	29.87	45.34

^1^Capsule-NLU model [15].

## Data Availability

All the datasets employed in the evaluations and the code are available at: https://github.com/keg-tucn/home-assist-capsnet accessed on 10 December 2020.

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
