# Peer review of "Intent Detection and Slot Filling with Capsule Net Architectures for a Romanian Home Assistant"

_sensors, 2021, doi:10.3390/s21041230_

Round 1

Reviewer 1 Report

The main idea of this paper is about intent detection and slot filling for a Romanian home assistant scenario. The proposed method is based on Capsule Neural Networks. Several problems in this paper are presented as follows:

1) The proposed SEMCAPSNET architecture only combines CAPSULE-NLU and INTENTCAPSNET, without much innovative work.

2) The quantitative result analysis of this paper is insufficient to illustrate the performance of the proposed method. The proposed model is just compared with Rasa model.

Author Response

Thank you for your review and feedback.

  1. The focus of the paper was on trying to understand the limitations introduced by the language challenges considered in the specific use case and on assessing if/how the proposed architecture, which combines the two mechanisms, can address them appropriately.
  2. In the evaluation part, the main focus was on discovering whether the language and learning challenges of the specific use case impose limitations on the performance of the architecture (what kind of limitations, how severe they are, how difficult to overcome, etc). Moreover, we considered that a critical, qualitative evaluation, in which we tried to understand the limitations of a certain strategy, can be more relevant from a research perspective, than a quantitative, comparative evaluation. We have previously considered Wit.ai in our evaluations, but removed it from further analysis due to its incapacity to handle diacritics. Also, RASA NLU employs CRF models, which still offer state-of-the-art performance for sequence tagging tasks. Comparison with more recent architectures, such as transformers, is a good suggestion, and we plan to consider it in our future work.

Reviewer 2 Report

The article applies a Capsule Net architecture for interpreting the statements for a Romanian home assistant. The intent detection and slot filling are evaluated. The manuscript is well written and represents a pleasant reading.

There are many results presented. Still, there could be some improvements that could make this article even more appealing, in my opinion.

Please briefly present the IOB labelling format in order to make the article more self-contained.

It would be very useful to make both the data set and the source code available. This would not only make the results repeatable, but they could serve for other researchers to take the study further.

It is not clear how are the confidence levels in subsection 5.4.3 computed. Please extend the description from lines 446-454 more. It would probably help to also provide some intuitive examples.

The values on the main diagonal in Figure 6 would be helpful. The reader is certainly curious to see how confident is the model on the samples that were correctly classified. Also it should be mentioned that the values in the confusion matrix represent percentages (this is what I understood), instead of the “normalized” term from the plot title. In general, I would discourage the plot titles and encourage the authors to put the details in the caption of the figure.

Line 514: “… values obtained from the attention matrix A.” Please be more precise.

Line 568 from the conclusions is maybe too strong: better results than the models trained with Rasa NLU are obtained in certain cases, as seen in Table 2.

Author Response

Thank you for your review, feedback and suggestions. We found them relevant, and we hope the paper is more valuable and better readable with the updates generated by your suggestions. We have included all of them in the new version of the paper (marked in red).

Reviewer 3 Report

The idea of this article and your research is interesting and useful for you and other authors who deal with Capsule Net Architectures. 

The figures and tables with the results of your research can be a little clearer and more precise for the readers.

Author Response

Thank you for your review and suggestions. In the next version of the paper we improved the quality of the images wherever the space permitted.